computational chemistry/green chemistry/ spectroscopy

troxerutin, calcium dobesilate, hydroquinone, synchronous fluorimetry, human plasma, greenness

**Author for correspondence:**
M. M. Salim
e-mail: mmasalim@mans.edu.eg

This article has been edited by the Royal Society of Chemistry, including the commissioning, peer review process and editorial aspects up to the point of acceptance.

# Simultaneous estimation of troxerutin and calcium dobesilate in presence of the carcinogenic hydroquinone using green spectrofluorimetric method

## M. M. Tolba[1], M. M. Salim[1,2] and M. El-Awady[1]

[1]Department of Pharmaceutical Analytical Chemistry, Faculty of Pharmacy, Mansoura University, Mansoura 35516, Egypt
[2]Department of Pharmaceutical Chemistry, Faculty of Pharmacy, Horus University-Egypt, New Damietta 34518, Egypt

MMT, 0000-0001-7968-6487; MMS, 0000-0003-4429-6504; ME-A, 0000-0003-2675-7530

In the present study, we conducted two facile and highly sensitive spectrofluorimetric approaches in order to quantify the vasoprotective agents; troxerutin (TROX) and calcium dobesilate (DOB) in the presence of hydroquinone (HQ) (as a highly toxic impurity and potential degradation product of DOB) in commercial formulations and human plasma. The first approach relies simply on using ethanol as an eco-friendly solvent for the estimation of DOB at 345 nm after being excited at 305 nm. The linearity was carefully investigated between DOB concentration and the relative fluorescence intensity in the range of 0.05–0.8 µg ml⁻¹. Due to the high method simplicity and sensitivity, applying the first approach to quality control analysis and spiked human plasma samples with mean % recoveries 100.74 ± 3.71 adds another merit. The second approach involved rapid conventional fluorimetric estimation of ethanolic TROX solution in TROX/DOB combined dosage forms at 455/350 nm (emission/excitation) with a linear calibration chart covering the range of 0.1–1.2 µg ml⁻¹. Moreover, the second approach involved a comprehensive study in a trial to solve the problem of superposition of DOB and HQ graph adopting the first derivative synchronous fluorimetric mechanism in ethanol at $\Delta\lambda = 60$ nm. Therefore, DOB was measured at 286 and 323 nm, while HQ could be quantitated at 301 nm. The Beer–Lambert Law has complied over the ranges of 0.1–1.0 and 0.02–0.4 µg ml⁻¹ for DOB and HQ, respectively. Guidelines adopted by the International

Council of Harmonization (ICH) were used to validate the target approaches. The developed methods are more convenient for routine quality control laboratory instead of the time-consuming and sophisticated reported techniques. Moreover, different aspects of evaluating the greenness of the proposed approaches were conducted to have a complete image of their environmental impact.

## 1. Introduction

Troxerutin (TROX, figure 1) is a trihydroxyethylrutoside; 3′,4′,7-Tris[O-(2-hydroxyethyl)]rutin [1]. TROX acts primarily to recover capillary function by decreasing the hyperpermeability and friability of micrangium. It is commonly used for preventing thrombosis and angiosclerosis [1].

Calcium dobesilate (DOB, figure 1) is a vasoprotective agent claimed to reduce capillary permeability. It may be used orally for the management of diabetic retinopathy or rectally for haemorrhoids [1]. Chemically, it is calcium di(2,5-dihydroxybenzenesulfonate) monohydrate [2]. DOB is officially registered in the British Pharmacopeia (BP) [2].

A combination of anti-haemorrhoidal, vasoprotective agents, DOB and TROX shows higher efficiency and better improvement of complaints in the treatment of varicose veins, diabetic retinopathy, hyper platelet aggregation, chronic venous insufficiency and haemorrhoids [3].

Numerous analytical techniques were reported in the literature for the estimation of TROX and DOB. For instance, TROX was estimated by spectrophotometry-based method [4], HPLC [5–9], electrochemical methods [10,11] and capillary electrophoresis [12]. On the other hand, DOB could be determined by spectrophotometric methods [13,14], HPLC [13,15–18], electrochemical methods [19–21] and chemiluminescence [22].

It is worth mentioning that hydroquinone (HQ, figure 1) is the DOB degradation product, resulting from its hydrolysis under acidic conditions. Moreover, it was incorporated in DOB monograph as its main impurity [2,17]. HQ's undesirable side effects that threaten the human being's life were monitored, such as nephrotoxicity, carcinogenicity and harmful impacts on the skin and eye [23,24]. By referring to the literature, only three methods were described as stability-indicating ones for estimation of DOB in the presence of HQ [16–18]. The previously reported methods suffered from different disadvantages, as some of them may not offer a sufficient quantification limit of the HQ. Moreover, all of the previous reports require expensive reagents and equipment [16–18].

A look at the literature revealed that no spectrofluorimetric approach was represented for estimation of TROX or DOB alone or concurrently in their combined dosage forms. Moreover, there are no described spectrofluorimetric techniques for parallel analysis of DOB in the presence of its highly toxic potential degradation product HQ. Synchronous fluorimetry either alone or combined with derivative spectrofluorimetry was used for the determination and separation of some drugs [25,26].

## 2. Experimental

### 2.1. Apparatus

— Experimental study was performed using quartz cell (1 cm), and Cary Eclipse fluorescence spectrophotometer (product of Agilent Technologies, USA) provided with xenon flash lamp superior sensitivity. The smoothing factor was adjusted to be 19.0, and the applied voltage was high. Scanning occurred in the range of 200–600 nm. Manipulation of the data was performed using Cary Eclipse software.
— Consort NV P-901 pH-meter (Belgium), IVM-300p vortex mixer (Taiwan) and 2–16P centrifuge (Germany) were used throughout the study.

### 2.2. Materials and reagents

— An authentic powder of TROX was obtained from Minapharm (Egypt) with a certified purity of 99.20%.
— Memphis Chemical Corporation (Cairo, Egypt) was the supplier for calcium dobesilate (with a labelled purity of 99.80%).
— Doxium 500 mg capsules, labelled to contain DOB (500 mg), batch no. 311131, was purchased from the local Egyptian pharmacy. It is a product of Memphis Corporation for Pharmaceutical and Chemical Industries (Egypt).

**Figure 1.** The structural formulae for TROX, DOB and HQ.

— Rutin tablets; labelled content of DOB/TROX is 500 mg each per tablet; batch no. 14TRU010 was purchased from the online market. It is a product of Endeavour, a division of Lupin Ltd, India.

— HPLC grade solvents were purchased from Fisher Scientific UK (UK), including methanol, ethanol and acetonitrile. While cetyltrimethylammonium chloride (CTAB; 99%) and sodium dodecyl sulfate (SDS; 95%) were produced by Sigma Aldrich (Germany).

— El-Nasr Pharmaceutical Chemicals Company (ADWIC) (Abu Zaabal, Egypt) was the supplier of HQ (with a certified purity of 98.30%), methylcellulose (MC), and tween-80. On the other hand, sodium acetate trihydrate, sodium hydroxide, acetic acid 96%, boric acid and hydrochloric acid (32%) were obtained from Winlab (UK).

— Mansoura University Hospitals (Mansoura, Egypt) was the supplier of human plasma, which was kept frozen until use after gentle thawing.

## 2.3. Standard solutions

Ten milligrams of the authentic powders of TROX, DOB and HQ were accurately weighed and transferred into 100.0 ml volumetric flask and dissolved in ethanol giving stock solutions equivalent to 100.0 µg ml$^{-1}$. Preparation of other respective standard solutions equivalent to 10, 5 and 2 µg ml$^{-1}$ for TROX, DOB and HQ was achieved by transferring suitable volumes into other 100.0 ml volumetric flasks completing with ethanol. After keeping all the standard solutions in the refrigerator at −2°C and monitoring the stability for 7 days via measuring its fluorescence intensities, no notable degradation was observed. The standard solutions could be used throughout this time.

## 2.4. Procedures

### 2.4.1. General procedures for calibration graphs

#### 2.4.1.1. First approach

The calibration graph in the range of 0.05–0.8 µg ml$^{-1}$ of DOB was constructed by accurate measurement and quantitative transfer of appropriate volumes of DOB working standard solutions into 10 ml

**Table 1.** Analytical performance data for the proposed approaches.

| parameter | the first approach (DOB) at 345 nm | the second approach | | | |
| --- | --- | --- | --- | --- | --- |
| | | TROX at 455 nm | DOB at 286 nm | DOB at 323 nm | HQ at 301 nm |
| linearity range (µg ml$^{-1}$) | 0.05–0.8 | 0.1–1.2 | 0.1–1.0 | 0.1–1.0 | 0.02–0.40 |
| intercept ($a$) | 30.749 | 20.466 | −0.290 | 0.818 | 0.916 |
| slope ($b$) | 854.00 | 516.218 | 28.04 | 16.54 | 98.54 |
| correlation coefficient ($r$) | 0.9999 | 0.9998 | 0.9999 | 0.9999 | 0.9999 |
| s.d. of residuals ($S_{y/x}$) | 2.74 | 4.99 | 0.13 | 0.09 | 0.16 |
| s.d. of intercept ($S_a$) | 1.85 | 3.59 | 0.10 | 0.07 | 0.11 |
| s.d. of slope ($S_b$) | 4.12 | 4.97 | 0.17 | 0.12 | 0.47 |
| percentage relative standard deviation, % RSD | 1.24 | 1.31 | 1.02 | 0.82 | 1.34 |
| percentage relative error, % error | 0.51 | 0.50 | 0.42 | 0.34 | 0.52 |
| limit of detection, LOD (ng ml$^{-1}$) | 7.00 | 22.90 | 12.00 | 14.00 | 3.50 |
| limit of quantitation, LOQ (ng ml$^{-1}$) | 21.70 | 69.50 | 37.00 | 43.70 | 10.60 |

volumetric flasks set, completing to the mark with ethanol and thoroughly mixing. A blank experiment was performed concurrently omitting the drug. Estimating the relative fluorescence intensities (RFI) of the resulted solutions was then achieved using emission maxima at 345 nm and excitation maxima at 305 nm.

### 2.4.1.2. Second approach

Aliquots of TROX standard solutions were moved quantitatively into a set of volumetric flasks (10 ml) to get solutions in the range of 0.1–1.2 µg ml$^{-1}$ final concentrations, after completing with ethanol. The prepared solutions' RFI were monitored at 455/350 nm, referring to emission/excitation, respectively.

For the second approach used for concurrent estimation of DOB and HQ with the existence of TROX, proper volumes of the corresponding DOB and HQ standard solutions were quantitatively transferred into two sets of calibrated 10-ml volumetric flasks. The flasks were completed to the mark with ethanol and mixed well to cover the final concentration range abridged in table 1. Synchronous spectra were recorded at a constant wavelength difference ($\Delta\lambda$) of 60 nm at a rate of 600 nm min$^{-1}$. Cary Eclipse software was used for the calculation of the first derivative amplitude of fluorescence spectra. Estimating the peak amplitude of the first derivative ($^1$D) spectra was performed at filter size 20 and interval 1.0 nm at 286 and 323 nm for DOB and 301 nm for HQ.

A concurrent blank experiment was carried out with the samples. The sample response (RFI or $^1$D derivative) was plotted against the concentration in µg ml$^{-1}$ to get the calibration graphs.

### 2.4.2. Analysis of TROX/DOB/HQ or DOB/HQ mixtures

TROX, DOB and HQ standard solution aliquots in various proportions cited in table 2 were accurately measured and placed into a set of calibrated 10 ml volumetric flasks. Then steps mentioned under 'General procedures for calibration graphs' were proceeded. TROX, DOB and HQ percentage recoveries were calculated either from the calibration graphs or using the regression equations.

**Table 2.** Application of the proposed approaches for the assessment of synthetic mixtures. N.B. each result is the average of three separate determinations.

| sample | amount taken (μg ml⁻¹) | | | amount found (μg ml⁻¹) | | | % found | | |
|---|---|---|---|---|---|---|---|---|---|
| | TROX | DOB | HQ | TROX | DOB at 286 nm | DOB at 323 nm | TROX | DOB at 286 nm | DOB at 323 nm |
| TROX, DOB and HQ synthetic mixture | 1.0 | 1.0 | 0.1 | 0.981 | 0.984 | 1.003 | 98.10 | 98.40 | 100.30 |
| | 0.8 | 0.8 | 0.08 | 0.809 | 0.796 | 0.794 | 101.13 | 99.50 | 99.25 |
| | 0.6 | 0.6 | 0.06 | 0.595 | 0.603 | 0.598 | 99.17 | 100.50 | 99.67 |
| | 0.4 | 0.4 | 0.04 | 0.392 | 0.391 | 0.401 | 98.00 | 97.75 | 100.25 |
| ($\bar{x}$) | | | | | | | 99.10 | 99.04 | 99.87 |
| ±s.d. | | | | | | | 1.26 | 1.21 | 0.50 |
| % RSD | | | | | | | 1.27 | 1.22 | 0.50 |
| % error | | | | | | | 0.64 | 0.61 | 0.25 |
| DOB and HQ synthetic mixture | | 1.0 | 0.02 | | 1.002 | 0.997 | | 100.20 | 99.70 |
| | | | 0.04 | | 0.993 | 0.987 | | 99.30 | 98.70 |
| | | | 0.06 | | 0.989 | 0.989 | | 98.90 | 98.90 |
| | | | 0.08 | | 1.006 | 0.994 | | 100.60 | 99.40 |
| | | | 0.1 | | 0.991 | 1.007 | | 99.10 | 100.70 |
| ($\bar{x}$) | | | | | | | | 99.62 | 99.48 |
| ±s.d. | | | | | | | | 0.74 | 0.79 |
| % RSD | | | | | | | | 0.74 | 0.79 |
| % error | | | | | | | | 0.33 | 0.35 |

### 2.4.3. Application of the proposed approaches for studied drugs estimation in pharmaceutical preparations

#### 2.4.3.1. Analysis of DOB in Doxium 500 mg capsule

An accurately weighed Doxium powder equivalent to 10.0 mg DOB was quantitatively transferred into a calibrated 100 ml volumetric flask. Ethanol (about 80 ml) was added, sonicated for 15 min then completed to 100 ml with the same solvent. The solutions were filtered, and aliquots were taken to analyse as the procedures under 'General procedures for calibration graphs'. The dosage form's nominal content was determined using the regression equation or the constructed calibration graph.

#### 2.4.3.2. Analysis of TROX and DOB simultaneously in Rutin tablets

Accurately, 10 tablets were weighed, ground and homogeneously mixed. Then 10.0 mg of TROX and DOB powdered tablets were put into a calibrated 100 ml volumetric flask. Ethanol (about 80 ml) was added, and the mixture content was sonicated for 30 min. The calibrated 100 ml volumetric flask was completed to the mark with ethanol and filtered (100 µg ml$^{-1}$ of each TROX and DOB). Accurately measured aliquots of the filtrate were analysed, as mentioned in 'General procedures for calibration graphs'. The dosage form's nominal content was determined either using the regression equation or from the constructed calibration graph.

### 2.4.4. Application of the first approach for DOB analysis in human plasma

One millilitre of human plasma was transferred into centrifugation tubes set and spiked with DOB standard solution so that the final DOB concentration was in the range of 0.1–0.8 µg ml$^{-1}$. The solutions were mixed well; 5.0 ml of ethanol was added to permit complete protein precipitation. The mixtures were subjected to vortex mixer for 5 min, followed by centrifuge at 3600 r.p.m. for 20 min at room temperature. One millilitre of the clear layer was transferred to a 10 ml calibrated measuring flask then completed to the mark with ethanol so that the final concentrations (0.1, 0.2, 0.4, 0.6 and 0.8 µg ml$^{-1}$ for DOB) were obtained. After that, the steps mentioned under 'General procedures for calibration graphs' were carried out in parallel with blank samples. The RFI was measured and plotted *against* the studied drug concentration in µg ml$^{-1}$.

# 3. Results and discussion

The absorption spectra of the ethanolic solution of DOB, TROX and HQ were scanned over the range 200–500 nm and the absorption reading at $\lambda_{max}$ 305, 350 and 295 nm was used for calculation of $\varepsilon$ ($8.2 \times 10^3$, $2.3 \times 10^4$ and $2.6 \times 10^3$ l mol$^{-1}$ cm$^{-1}$) of the studied compounds, respectively (electronic supplementary material, figure S1). From the absorption spectra data and the phenolic molecular structure of all the studied compounds, the excitation wavelengths were selected for the native fluorescence study.

DOB ethanolic solution showed intense emission/excitation bands at 345/305 nm (figure 2*a*). Due to the high sensitivity of the first approach, DOB could be assessed in its pharmaceutical capsules and spiked human plasma as an alternative to the reported complicated and time-consuming methods.

The native fluorescence spectra of TROX (second approach) were firmly shown in ethanol at 455 nm after excitation at 350 nm without interference from the co-formulated DOB (figure 2*a*). Figure 2*b* represents the emission spectra for increasing concentrations of TROX in ethanol at 455 nm after excitation at 350 nm in the presence of DOB and HQ. So, the second approach's suggested procedure was widely applicable in the quality control laboratories for analysis of the commercial co-formulated Rutin tablets.

After preliminary studies, both conventional and synchronous fluorimetric estimation could not separate DOB from the toxic impurity and potential degradation product HQ (figures 2 and 3). Accordingly, the proposed studies were devoted to first derivative synchronous fluorimetry (FDSF), which resolved DOB from its toxic impurity and main degradation product (HQ) in an efficient manner permitting its determination at both 286 and 323 nm, as illustrated in figure 4. Similarly, HQ was quantified at 301 nm after applying FDSF that detaches its band from its descent drug DOB (figure 5), proving the stability-indicating attitude of this method. Moreover, a straightforward green approach was established for limit estimation of the carcinogenic and nephrotoxic HQ in DOB raw materials and pharmaceutical formulations, as shown in figure 6.

## 3.1. Optimization of experimental conditions

The parameters affecting the studied compounds' fluorescence intensities were experimentally optimized to obtain the ultimate selectivity and the highest sensitivity.

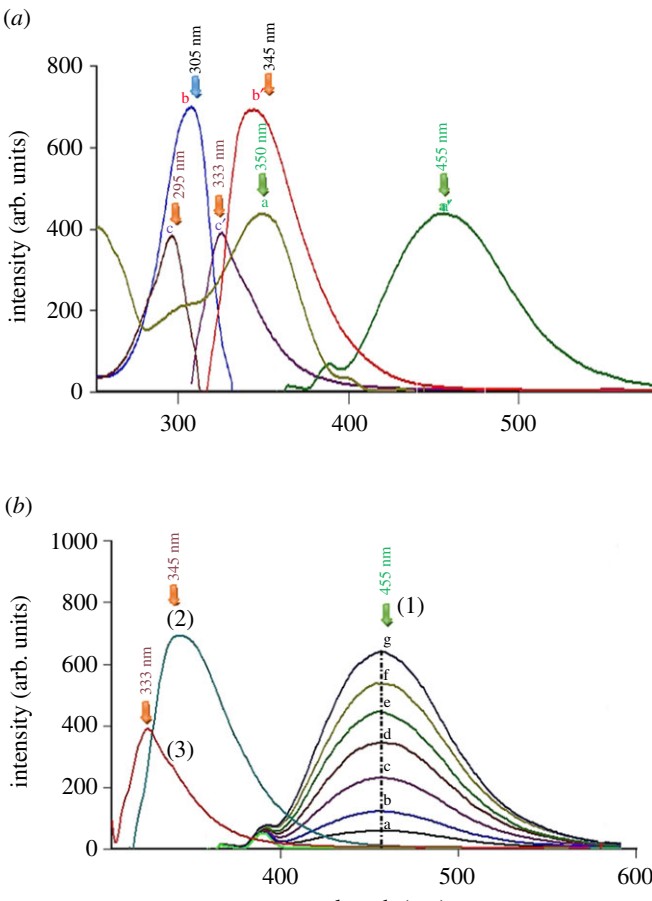

**Figure 2.** (*a*) Excitation and emission fluorescence spectra of a, a′ TROX (0.8 µg ml$^{-1}$) b, b′ DOB (0.8 µg ml$^{-1}$) c, c′ HQ (0.08 µg ml$^{-1}$) and (*b*) emission fluorescence spectra of (1) (a–g) of TROX (0.1, 0.2, 0.4, 0.6, 0.8, 1 and 1.2 µg ml$^{-1}$) at 455 nm, (2) DOB (0.8 µg ml$^{-1}$) and (3) HQ (0.08 µg ml$^{-1}$).

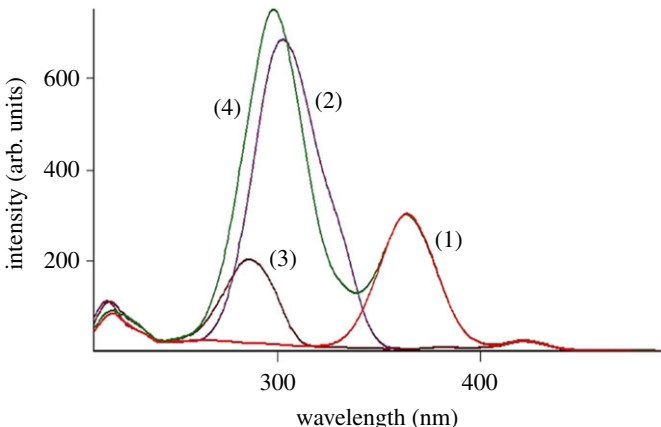

**Figure 3.** Synchronous fluorescence spectra of (1) TROX (1 µg ml$^{-1}$), (2) DOB (1 µg ml$^{-1}$), (3) HQ (0.1 µg ml$^{-1}$) and (4) synthetic mixture of TROX (1 µg ml$^{-1}$), DOB (1 µg ml$^{-1}$) and HQ (0.1 µg ml$^{-1}$).

### 3.1.1. Optimization of diluting solvent

The impact of different diluting solvents on the aforementioned analytes' fluorescence intensities was conducted using methanol, ethanol, water, acetonitrile and n-propanol. As illustrated in figure 7, ethanol gave the highest fluorescence intensity of TROX, so; it was the best diluent for its determination. It is worth noting that the RFI values of DOB and HQ were very close in methanol, ethanol and acetonitrile; however, ethanol greenness encourages its use. At the emission wavelengths

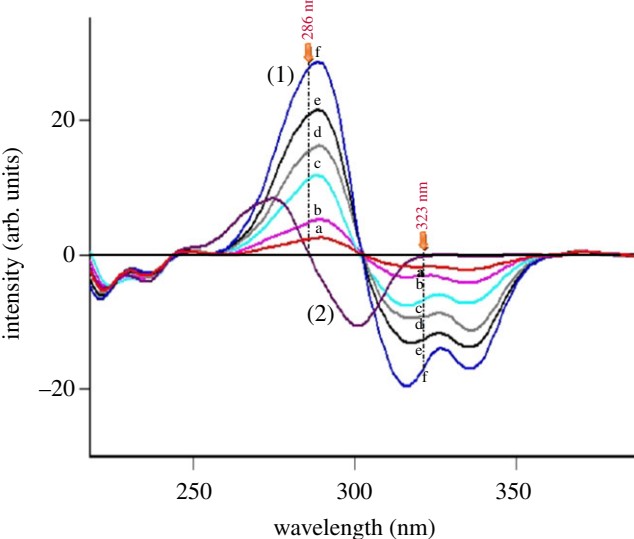

**Figure 4.** First derivative synchronous fluorescence spectra of (1) (a–f) DOB (0.1, 0.2, 0.4, 0.6, 0.8 and 1 μg ml$^{-1}$) at 286 nm and at 323 nm and (2) HQ (0.1 μg ml$^{-1}$).

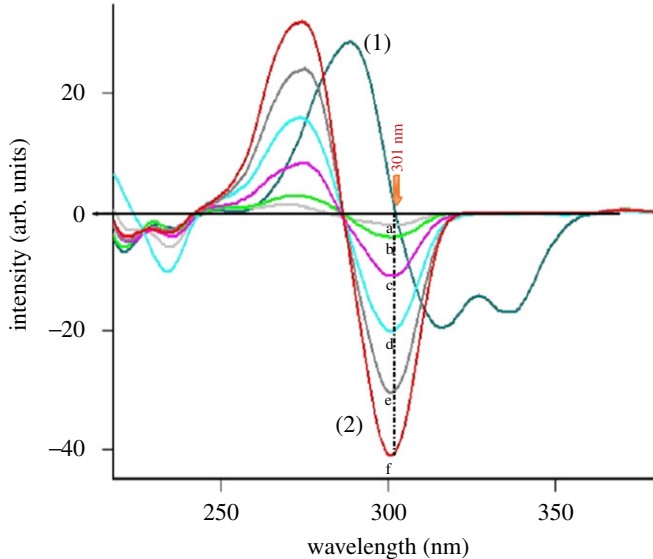

**Figure 5.** First derivative synchronous fluorescence spectra of (1) DOB (1 μg ml$^{-1}$) and (2) (a–f) of HQ (0.02, 0.04, 0.1, 0.2, 0.3 and 0.4 μg ml$^{-1}$) at 301 nm.

of both DOB and HQ, a high reference reading was recorded using n-propanol. While using water resulted in a notable decrease in the fluorescence intensity of either HQ or TROX. So, ethanol was selected as the best green solvent achieving adequate sensitivity.

### 3.1.2. Optimization of the medium pH

Various types of buffers were checked to evaluate the influence of pH on the analytes' fluorescence intensities. The acidic pH range was covered using 0.1 M HCl and 0.2 M acetate buffer (pH 3–5), while 0.2 M borate buffer (pH 6–9) and 0.1 M NaOH covering the basic pH range were tested. As shown in electronic supplementary material, figure S2, it was noted that increasing the pH (2–9) did not remarkably enhance the RFI of the analytes. At the same time, using 0.1 M NaOH as a pH modifier resulted in a decrease in RFI of TROX and quenching of the RFI of DOB and HQ, which could be attributed to their degradation. Conversely, ethanol only increases the RFI of the studied compounds; therefore, completing the volume with ethanol was chosen as the optimum condition.

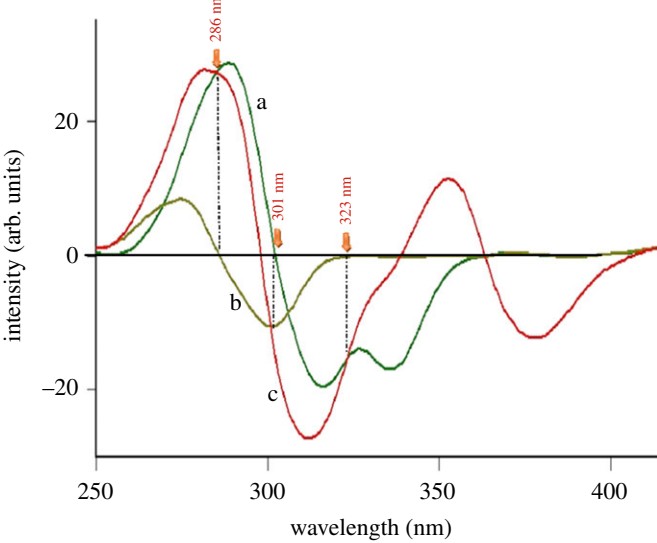

**Figure 6.** First derivative synchronous fluorescence spectra of (a) 1 µg ml$^{-1}$ DOB, (b) 0.1 µg ml$^{-1}$ HQ and (c) mixture of 1 µg ml$^{-1}$ TROX, 1 µg ml$^{-1}$ DOB and 0.1 µg ml$^{-1}$ HQ.

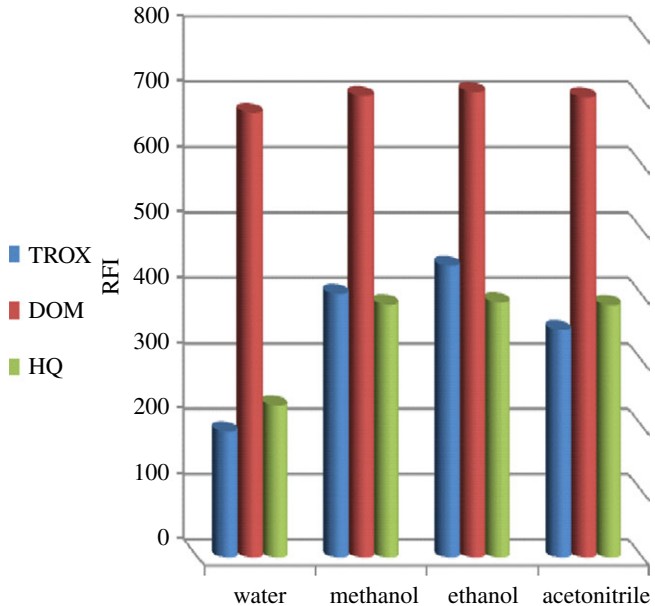

**Figure 7.** Effect of different solvent on the native fluorescence intensity of TROX (0.8 µg ml$^{-1}$), DOB (0.8 µg ml$^{-1}$) and HQ (0.08 µg ml$^{-1}$).

### 3.1.3. Effect of surfactant

The surfactant effect was accomplished using aqueous solutions (0.5% w/v) of a non-ionic surfactant (tween-80), anionic surfactant (SDS), cationic surfactant (CTAB) and macromolecules (methylcellulose). As illustrated in electronic supplementary material, figure S3, the studied compounds' fluorescence intensities were not affected by these surfactants. Consequently, there is no need for their incorporation in the procedure.

### 3.1.4. Optimum $\Delta\lambda$ effect in the second approach

Optimization of $\Delta\lambda$ is an essential criterion that should be considered, as it may significantly affect resolution, sensitivity and band symmetry. Consequently, an investigation of $\Delta\lambda$ over a wide range of (20–140 nm) was made. Adequate sensitivity and suitable band shapes were obtained upon

using $\Delta\lambda$ of 60. Higher and lower values of $\Delta\lambda$ than 60 showed low fluorescence intensity for the studied compounds.

## 3.2. Validation of the approaches

### 3.2.1. Linearity and range

A rectilinear relationship was constructed by graphing either the RFI or the peak amplitude ($^1$D) against the studied drug concentration, and concentration ranges for each drug by the two adopted approaches are cited in table 1. Statistical analysis of the obtained data gave the following regression equations:

#### 3.2.1.1. First approach

$$\mathrm{RFI} = 30.749 + 854\mathrm{C} \quad (r = 0.9999) \quad \text{for DOB at 345 nm.}$$

#### 3.2.1.2. Second approach

$$
\begin{array}{llll}
\mathrm{RFI} = 20.466 + 516.218\ \mathrm{C} & (r = 0.9998) & \text{for TROX at 455 nm,} \\
1\mathrm{D} = -0.290 + 28.04\ \mathrm{C} & (r = 0.9999) & \text{for DOB at 286 nm,} \\
1\mathrm{D} = 0.818 + 16.54\ \mathrm{C} & (r = 0.9999) & \text{for DOB at 323 nm} \\
\text{and} \quad 1\mathrm{D} = 0.916 + 98.54\ \mathrm{C} & (r = 0.9999) & \text{for HQ at 301 nm,}
\end{array}
$$

where RFI is the relative fluorescence intensity, C is the drug's concentration in µg ml$^{-1}$, $^1$D is the peak amplitude in the first derivative synchronous fluorescence technique, and $r$ is the correlation coefficient.

Statistical analysis [27] of the obtained data proved the calibration graph's linearity, as summarized in table 1.

### 3.2.2. Quantitation limit (LOQ) and detection limit (LOD)

These analytical parameters were computed mathematically by the equations specified by ICH guidelines [28] and were abridged in table 1

$$\mathrm{LOQ} = 10\ S_a/b \quad \text{and} \quad \mathrm{LOD} = 3.3\ S_a/b.$$

where $S_a$ = standard deviation of the intercept of the calibration curve and $b$ = slope of the calibration curve.

### 3.2.3. Selectivity

The selectivity tests whether the analytical method could unequivocally assess the analyte in the presence of expected components or not [28]. The proposed approaches' selectivity was assessed using synthetic mixtures of TROX, DOB and HQ (table 2). Moreover, it was also confirmed by the determination of TROX and DOB in different commercial pharmaceutical preparations (Rutin tablets and Doxium capsules) without common excipients interferences. The two approaches were specific for DOB in biological samples, as no interferences from the endogenous plasma proteins were observed. Besides, the second approach proved to be selective for DOB in the presence of its toxic impurity and degradation product HQ.

### 3.2.4. Precision and accuracy

The repeatability and intermediate precision show small RSD values indicating high precision of the proposed analytical approaches (table 3). The accuracy of the proposed analytical approaches was also assessed via statistical analysis comparing between the obtained results and comparison method [6], as shown in table 4. No significant difference was found using Student's $t$-test and variance ratio $F$-test [27]. Moreover, the acceptable percentage recoveries demonstrated the accuracy of the proposed analytical approaches over the specified range as abridged in table 4.

**Table 3.** Precision data for the assessment of the studied drugs by the proposed approaches.

| amount taken ($\mu$g ml$^{-1}$) | % found | % RSD | % error |
|---|---|---|---|
| the first approach (DOB) | | | |
| intraday; 0.4 | 99.17 ± 0.56 | 0.56 | 0.33 |
| 0.6 | 99.34 ± 0.34 | 0.34 | 0.20 |
| 0.8 | 100.15 ± 0.67 | 0.67 | 0.39 |
| interday; 0.4 | 101.11 ± 1.02 | 1.01 | 0.58 |
| 0.6 | 99.91 ± 0.92 | 0.92 | 0.53 |
| 0.8 | 100.55 ± 1.21 | 1.20 | 0.69 |
| the second approach | | | |
| TROX at 455 nm | | | |
| intraday; 0.6 | 100.37 ± 0.72 | 0.72 | 0.41 |
| 0.8 | 98.93 ± 0.51 | 0.52 | 0.30 |
| 1.0 | 99.76 ± 0.90 | 0.90 | 0.52 |
| interday; 0.6 | 99.18 ± 1.3 | 1.31 | 0.76 |
| 0.8 | 98.83 ± 0.70 | 0.71 | 0.41 |
| 1.0 | 99.37 ± 1.05 | 1.06 | 0.61 |
| DOB at 286 nm | | | |
| intraday; 0.4 | 100.01 ± 0.82 | 0.82 | 0.47 |
| 0.6 | 99.92 ± 0.75 | 0.75 | 0.43 |
| 0.8 | 99.11 ± 0.66 | 0.67 | 0.38 |
| interday; 0.4 | 98.71 ± 0.41 | 0.42 | 0.24 |
| 0.6 | 97.81 ± 0.65 | 0.66 | 0.38 |
| 0.8 | 99.02 ± 0.91 | 0.92 | 0.53 |
| DOB at 323 nm | | | |
| Intraday; 0.4 | 99.13 ± 0.34 | 0.34 | 0.20 |
| 0.6 | 101.05 ± 0.73 | 0.72 | 0.42 |
| 0.8 | 100.22 ± 0.45 | 0.45 | 0.26 |
| interday; 0.4 | 101.73 ± 1.51 | 1.48 | 0.85 |
| 0.6 | 99.63 ± 0.95 | 0.95 | 0.55 |
| 0.8 | 101.11 ± 1.36 | 1.35 | 0.78 |

## 3.3. Applications

### 3.3.1. Analysis of TROX/DOB/HQ or DOB/HQ synthetic mixtures

The analytical fluorimetric approaches were investigated to determine TROX, DOB and HQ simultaneously in the synthetic mixtures with varied ratios (second approach). The linear regression equations calculated the concentrations of each compound. The results point out the high accuracy of the proposed analytical approach (table 2).

### 3.3.2. Application of the analytical proposed approaches in pharmaceutical dosage forms

The proposed procedure was applied successfully to estimate DOB in Doxium capsules and TROX/DOB in the pharmaceutical co-formulated Rutin tablets using the first and second approaches, respectively. Satisfactory results were obtained for DOB and TROX/DOB, which agreed with the label claims (table 4).

Statistical analysis [27] of the results obtained by comparing the proposed approaches and comparison method [6] proved no significant difference between their performance with regards to the accuracy and precision, respectively.

**Table 4.** Application of the proposed approaches for the assessment of the pharmaceutical dosage forms. N.B. Each result is the average of three separate determinations.

| sample | amount taken ($\mu$g ml$^{-1}$) | amount found ($\mu$g ml$^{-1}$) | | | % found | | | comparison method [6] % found |
|---|---|---|---|---|---|---|---|---|
| | DOB | DOB at 345 nm | DOB at 286 nm | DOB at 323 nm | DOB at 345 nm | DOB at 286 nm | DOB at 323 nm | DOB |
| Doxium 500 mg capsules | 0.4 | 0.398 | 0.398 | 0.401 | 99.50 | 99.50 | 100.25 | 102.00 |
| (500 mg DOB/capsule) | 0.6 | 0.599 | 0.600 | 0.606 | 99.80 | 100.00 | 101.00 | 100.63 |
| first and second approach | 0.8 | 0.801 | 0.797 | 0.799 | 100.13 | 99.63 | 99.88 | 99.80 |
| $\bar{x} \pm$ s.d. | | | | | 99.81 ± 0.32 | 99.71 ± 0.26 | 99.71 ± 0.57 | 100.81 ± 1.11 |
| t | | | | | 1.50 | 1.67 | 0.60 | |
| F | | | | | 12.43 | 18.34 | 3.79 | |
| nominal content | | | | | 499.05 | 498.55 | 498.55 | |

| sample | amount taken ($\mu$g ml$^{-1}$) | | amount found ($\mu$g ml$^{-1}$) | | | % found | | | comparison method [6] % found | |
|---|---|---|---|---|---|---|---|---|---|---|
| | TROX | DOB | TROX | DOB at 286 nm | DOB at 323 nm | TROX | DOB at 286 nm | DOB at 323 nm | TROX | DOB |
| Rutin tablets | 1.0 | 1.0 | 0.997 | 1.002 | 0.999 | 99.70 | 100.20 | 99.90 | 99.75 | 100.35 |
| (500 mg TROX and 500 mg DOB) | 0.8 | 0.8 | 0.792 | 0.795 | 0.793 | 99.00 | 99.38 | 99.13 | 97.86 | 101.42 |
| second approach | 0.6 | 0.6 | 0.601 | 0.599 | 0.594 | 100.17 | 99.83 | 99.00 | 99.32 | 99.96 |
| $\bar{x} \pm$ s.d. | | | | | | 99.62 ± 0.59 | 99.80 ± 0.41 | 99.34 ± 0.49 | 100.48 ± 0.99 | 100.58 ± 0.76 |
| t | | | | | | 0.97 | 1.56 | 2.37 | | |
| F | | | | | | 2.38 | 3.39 | 2.41 | | |
| nominal content | | | | | | 498.10 | 499.00 | 496.70 | | |

[a]The tabulated t- and F-values are 2.78 and 19, respectively at p = 0.05 [27].

**Table 5.** Assay results for the determination of calcium dobesilate in spiked human plasma samples using the first approach.

| parameter | amount taken ($\mu$g ml$^{-1}$) | amount found ($\mu$g ml$^{-1}$) | % found |
|---|---|---|---|
| spiked human plasma | 0.10 | 0.107 | 107.00 |
| | 0.20 | 0.197 | 98.50 |
| | 0.40 | 0.390 | 97.50 |
| | 0.60 | 0.602 | 100.33 |
| | 0.80 | 0.803 | 100.38 |
| $\bar{x}$ | | | 100.74 |
| $\pm$s.d. | | | $\pm$3.71 |
| % RSD | | | 3.68 |
| % error | | | 1.65 |

### 3.3.3. Application of the proposed first approach to the determination of DOB in human plasma

After oral administration of Doxium capsules (500 mg of calcium dobesilate), the maximum concentration ($C_{max}$) was an average of 8 $\mu$g ml$^{-1}$ after 6 h [29]. The method's sensitivity permitted DOB determination in spiked human plasma following a simple protein precipitation method. Electronic supplementary material, figure S4 shows DOB determination (0.6 $\mu$g ml$^{-1}$) in spiked human plasma using the first approach. Table 5 summarizes the results of the determination of DOB in spiked human plasma using the first fluorimetric approach. Consequently, it is expected to be applied for the therapeutic drug monitoring of DOB in real human plasma samples with promising results.

## 3.4. Evaluation of the environmental impact of the proposed approaches

The replacement of toxic solvents and reagents with others having lower toxicity is an essential step for the analytical method to have eco-friendly properties. Considering the proposed analytical methods, they were found to be benign to the environment. Different analytical tools are well known in this field as the national environmental methods index (NEMI) [30], analytical Eco-scale [31] and green analytical procedure index (GAPI) [32]. The three mentioned tools have been applied to our proposed methods in order to assess its greenness. The estimation using the three tools is summarized as follows:

— NEMI: It is a graph divided into four quarters. The green colour indicates that the procedure's reagents are not hazardous, persistently bio-accumulative and toxic or corrosive or produce no significant amounts of wastes. As the proposed methods using ethanol in the preparation of samples and as diluting solvents, the four quarters coloured green (table 6).
— Eco-scale: which depends on penalty points as evaluation. The initial score for the method is 100 points. If it deviates from the ideal one, the penalty points are subtracted from the base value. For the proposed methods, the Eco-scale score = 89 (table 6).
— GAPI: New aspects have been incorporated here to get a complete image of the developed method. It is presented by five pentagrams to quantify the impact on the environment. The items are coloured green, yellow or red, pointing out low, medium and high environmental impacts. The pictogram for the proposed method is presented in table 6.

The overall conclusion from this visual representation is that our proposed methods comply with the aforementioned tools' greenness parameters to a great extent due to their minimal impacts on human health and the environment.

# 4. Conclusion

The proposed fluorimetric approaches provide fully validated, accurate and straightforward methods for the quantitative assessment of DOB either alone or simultaneously with TROX and/or HQ. The proposed approaches were used to successfully determine TROX and DOB in its pharmaceutical co-formulation as an alternative to the complicated HPLC reports or the less-sensitive

**Table 6.** Results for evaluation of greenness of the proposed approaches.

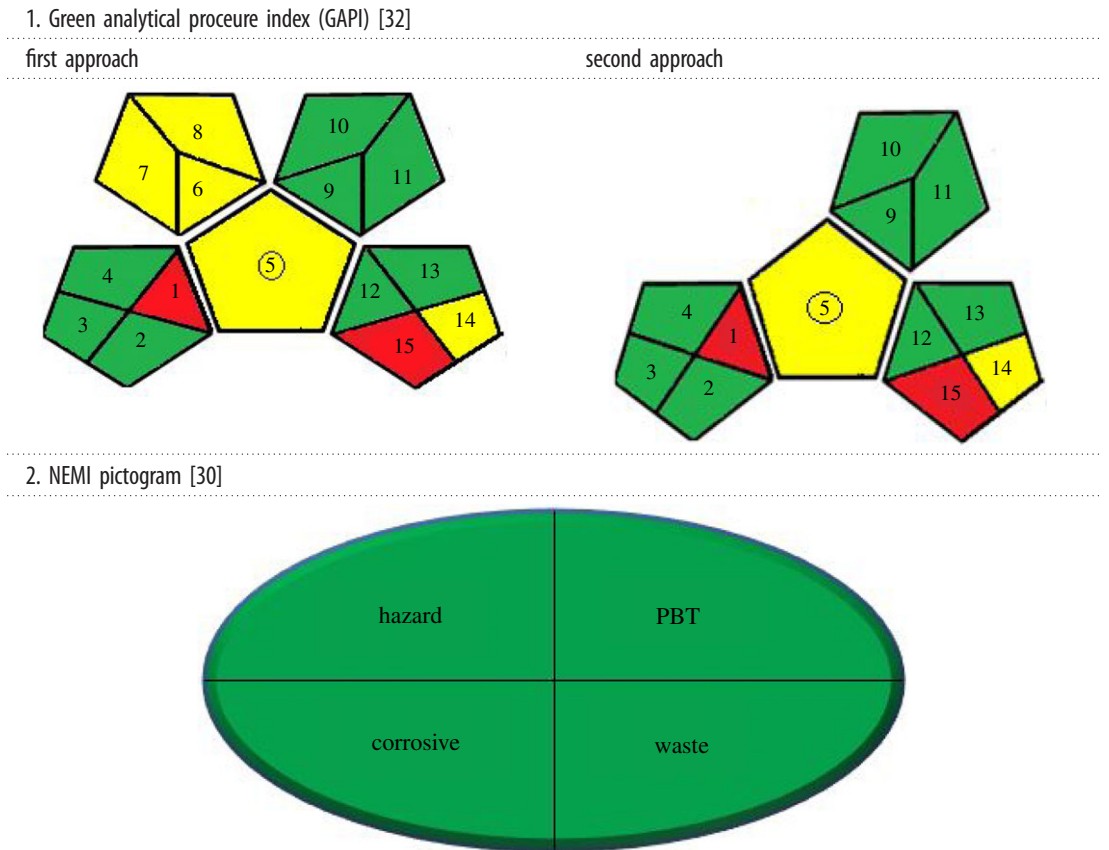

### 1. Green analytical proceure index (GAPI) [32]

first approach                    second approach

### 2. NEMI pictogram [30]

### 3. Analytical Eco-scale score [31]

| item | no. of pictogram | word sign | penalty points |
| --- | --- | --- | --- |
| — reagent; volume (ml) | 2 | danger | 8 |
| ethanol; 10 ml | | | |
| — spectrofluorimeter | | | 0 |
| — occupational hazard | | | 0 |
| — waste | | | 3 |
| — total penalty points | | | 11 |
| — analytical Eco-scale | | | 89 |
| score | | | |

spectrophotometric ones. The proposed spectrofluorimetric approaches were ideal for determining DOB in human plasma with no need for a tedious plasma extraction procedure. An excellent resolution was achieved for the DOB band from its main toxic degradation product HQ using $\Delta\lambda$ of 60 nm by the first derivative synchronous fluorimetry. Hence, the stability-indicating capability of this approach was proven. At first glance, the proposed methods seem to have a benign ecological impact as they coincide with all the requirements of NEMI's greenness tools, analytical Eco-scale and GAPI.

Data accessibility. Data are available from the Dryad Digital Repository: https://doi.org/10.5061/dryad.dv41ns1w9 [33].
Authors' contributions. M.M.T. carried out the laboratory work, participated in data analysis and participated in the design of the study; M.M.S. drafted the manuscript, carried out the statistical analyses, conceived of the study and designed the study; M.E.-A. coordinated the study participated in data analysis and helped draft the manuscript. All authors gave final approval for publication.
Competing interests. We declare we have no competing interests.
Funding. We received no funding for this study.

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

*Spectrochim. Acta A Mol. Biomol. Spectrosc.* **60**, 2377–2382. (doi:10.1016/j.saa.2003.12.011)

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
