## [Peer Review File · Royal Society Open Science]

Review History

RSOS-201888.R0 (Original submission)

Review form: Reviewer 1 (Mohamed Attwa)

Is the manuscript scientifically sound in its present form?

Yes

Are the interpretations and conclusions justified by the results?

Yes

Is the language acceptable?

Yes

Do you have any ethical concerns with this paper?

No

Have you any concerns about statistical analyses in this paper?

No

Recommendation?

Accept with minor revision (please list in comments)

Comments to the Author(s)

The manuscript presents two spectrofluorimetric methods for the determination of calcium dobesilate alone or simultaneously with troxerutin and its potential degradation product; hydroquinone. The current study is the first report for the spectrofluorimetric assay of this mixture. The work described is valuable and suitable for publication in Royal Society Open Science (RSOS) after considering the following revisions:

- (1) The title can be better rewritten in a clearer way as follows: Simultaneous estimation of troxerutin and calcium dobesilate in presence of the carcinogenic degradation product hydroquinone using green spectrofluorimetric method.
- (2) The manuscript includes some minor English mistakes and should be generally revised regarding grammatical or spelling mistakes.
- (3) P2, L25: Replace "Due to the method's high simplicity and sensitivity" by "Due to the high method simplicity and sensitivity".
- (4) Unify the use of the term "fluorimetry" instead of "fluorometry" all over the text.
- (5) P2, L30: Replace "TROX ethanolic solution" by "ethanolic TROX solution".
- (6) P2, L44: Replace "They were" by "The developed methods are".
- (7) Avoid writing details about the results at the introduction section.
- (8) Did the authors study the effect of surfactants in water or in ethanol?
- (9) P7, L3: Replace "Results and the discussion" by "Results and discussion".
- (10) In "Section 3. Results and discussion", please give a highlight to the relation of the chemical structures of the studied analytes and the presence of their native fluorescence.
- (11) P8, L4: Please re-phrase this sentence: "All the mentioned attributes forced us to continue using ethanol as diluent".
- (12) P11, L34: The sentence starting with "Considering the proposed" should be rewritten in a clear way.
- (13) The access dates in references 23, 26 and 27 should be updated and the URL addresses in reference 26 and 27 should be checked.

Also, it is preferable to include reference to royal society open science journal:

Ex:

- 1- Abdelhameed AS, Attwa MW, Kadi AA. Liquid chromatography–tandem mass spectrometry metabolic profiling of nazartinib reveals the formation of unexpected reactive metabolites. Royal Society Open Science. 2019;6(8):190852.
- 2- Abdelhameed AS, Attwa MW, Al-Shaklia NS, Kadi AA. A highly sensitive LC-MS/MS method to determine novel Bruton's tyrosine kinase inhibitor spebrutinib: application to metabolic stability evaluation. Royal Society Open Science. 2019;6(6):190434.
- 3- Kadi AA, Darwish HW, Abuelizz HA, Alsubi TA, Attwa MW. Identification of reactive intermediate formation and bioactivation pathways in Abemaciclib metabolism by LC–MS/MS: in vitro metabolic investigation. Royal Society Open Science. 2019;6(1):181714.

- (14) References corresponding to the methods used for evaluation of greenness should be added to Table 6.
- (15) The manuscript lacks the figure numbers on the presented figures.

Review form: Reviewer 2**Is the manuscript scientifically sound in its present form?**

Yes

Are the interpretations and conclusions justified by the results?

Yes

Is the language acceptable?

Yes

Do you have any ethical concerns with this paper?

No

Have you any concerns about statistical analyses in this paper?

No

Recommendation?

Accept with minor revision (please list in comments)

Comments to the Author(s)

I read the paper many times, it sounds acceptable.

One remark, the authors should give the values of the optical densities at the excitation and the emission wavelengths, used, to obtain excitation and emission spectra.

Second remark: They should write Fluorescence excitation and emission spectra.

Review form: Reviewer 3

Is the manuscript scientifically sound in its present form?

Yes

Are the interpretations and conclusions justified by the results?

Yes

Is the language acceptable?

Yes

Do you have any ethical concerns with this paper?

No

Have you any concerns about statistical analyses in this paper?

No

Recommendation?

Accept with minor revision (please list in comments)

Comments to the Author(s)

The manuscript# RSOS-201888 entitled "Green spectrofluorimetric estimation of troxerutin and dobesilate simultaneously in the presence of the carcinogenic degradation product hydroquinone" is of interest and suitable for the scope of Royal Society Open Science. This paper may be published after a revision considering the following points:

1. The language needs some revision and editing.
2. Page#2, Line#32: Remove the space at the end of the paragraph and make the abstract one paragraph.

3. Page#2, Line#46: Replace “evaluating the proposed approaches' greenness” by “evaluating the greenness of the proposed approaches”
4. The purity of the studied drugs [troxerutin, dobesilate, and hydroquinine] should be written.
5. Under the procedure section, the statement "completing to the volume" must be changed to be "completing to the mark."
6. Page #6 Line # 2: concentration in an ascending way.....should be deleted.
7. In section # 2.4.3.2.: The dosage form's nominal content was determined....., the obtained statistical data should be added in Table #4.
8. Page#7 Line #10: We used the higher sensitivity of the first approach..... Please rewrite the sentences in an acceptable scientific writing way.
9. Page#7, Line#42: Replace “Experimental condition optimization” by “Optimization of experimental conditions”.
10. Reference #14.....please, adhere to the journal guidelines for reference formatting.
11. The resolution and the shape of figure #1 are poor. Please, try to improve.
12. The number of figures is a large number and some of them can be moved to the supplementary material.

Decision letter (RSOS-201888.R0)

Dear Dr Salim:

Title: Green spectrofluorimetric estimation of troxerutin and dobesilate simultaneously in the presence of carcinogenic degradation product hydroquinone
Manuscript ID: RSOS-201888

The editor assigned to your manuscript has now received comments from reviewers. We would like you to revise your paper in accordance with the referee and Subject Editor suggestions which can be found below (not including confidential reports to the Editor). Please note this decision does not guarantee eventual acceptance.

Please submit your revised paper before 11-Dec-2020. Please note that the revision deadline will expire at 00.00am on this date. If we do not hear from you within this time then it will be assumed that the paper has been withdrawn. In exceptional circumstances, extensions may be possible if agreed with the Editorial Office in advance. We do not allow multiple rounds of revision so we urge you to make every effort to fully address all of the comments at this stage. If deemed necessary by the Editors, your manuscript will be sent back to one or more of the original reviewers for assessment. If the original reviewers are not available we may invite new reviewers.

To revise your manuscript, log into <http://mc.manuscriptcentral.com/rsos> and enter your Author Centre, where you will find your manuscript title listed under "Manuscripts with Decisions." Under "Actions," click on "Create a Revision." Your manuscript number has been

appended to denote a revision. Revise your manuscript and upload a new version through your Author Centre.

RSC Associate Editor:
Comments to the Author:
(There are no comments.)

RSC Subject Editor:
Comments to the Author:
(There are no comments.)

Reviewers' Comments to Author:
Reviewer: 1

Comments to the Author(s)

The manuscript presents two spectrofluorimetric methods for the determination of calcium dobesilate alone or simultaneously with troxerutin and its potential degradation product; hydroquinone. The current study is the first report for the spectrofluorimetric assay of this mixture. The work described is valuable and suitable for publication in Royal Society Open Science (RSOS) after considering the following revisions:

- (1) The title can be better rewritten in a clearer way as follows: Simultaneous estimation of troxerutin and calcium dobesilate in presence of the carcinogenic degradation product hydroquinone using green spectrofluorimetric method.
- (2) The manuscript includes some minor English mistakes and should be generally revised regarding grammatical or spelling mistakes.

- (3) P2, L25: Replace "Due to the method's high simplicity and sensitivity" by "Due to the high method simplicity and sensitivity".
- (4) Unify the use of the term "fluorimetry" instead of "fluorometry" all over the text.
- (5) P2, L30: Replace "TROX ethanolic solution" by "ethanolic TROX solution".
- (6) P2, L44: Replace "They were" by "The developed methods are".
- (7) Avoid writing details about the results at the introduction section.
- (8) Did the authors study the effect of surfactants in water or in ethanol?
- (9) P7, L3: Replace "Results and the discussion" by "Results and discussion".
- (10) In "Section 3. Results and discussion", please give a highlight to the relation of the chemical structures of the studied analytes and the presence of their native fluorescence.
- (11) P8, L4: Please re-phrase this sentence: "All the mentioned attributes forced us to continue using ethanol as diluent".
- (12) P11, L34: The sentence starting with "Considering the proposed" should be rewritten in a clear way.
- (13) The access dates in references 23, 26 and 27 should be updated and the URL addresses in reference 26 and 27 should be checked.

Also, it is preferable to include reference to royal society open science journal:

Ex:

- 1- Abdelhameed AS, Attwa MW, Kadi AA. Liquid chromatography–tandem mass spectrometry metabolic profiling of nazartinib reveals the formation of unexpected reactive metabolites. *Royal Society Open Science*. 2019;6(8):190852.
- 2- Abdelhameed AS, Attwa MW, Al-Shaklia NS, Kadi AA. A highly sensitive LC-MS/MS method to determine novel Bruton's tyrosine kinase inhibitor spebrutinib: application to metabolic stability evaluation. *Royal Society Open Science*. 2019;6(6):190434.
- 3- Kadi AA, Darwish HW, Abuelizz HA, Alsubi TA, Attwa MW. Identification of reactive intermediate formation and bioactivation pathways in Abemaciclib metabolism by LC-MS/MS: in vitro metabolic investigation. *Royal Society Open Science*. 2019;6(1):181714.

- (14) References corresponding to the methods used for evaluation of greenness should be added to Table 6.
- (15) The manuscript lacks the figure numbers on the presented figures.

Reviewer: 2

Comments to the Author(s)

I read the paper many times, it sounds acceptable.

One remark, the authors should give the values of the optical densities at the excitation and the emission wavelengths, used, to obtain excitation and emission spectra.

Second remark: They should write Fluorescence excitation and emission spectra.

Reviewer: 3

Comments to the Author(s)

The manuscript# RSOS-201888 entitled "Green spectrofluorimetric estimation of troxerutin and dobesilate simultaneously in the presence of the carcinogenic degradation product hydroquinone" is of interest and suitable for the scope of Royal Society Open Science. This paper may be published after a revision considering the following points:

1. The language needs some revision and editing.
2. Page#2, Line#32: Remove the space at the end of the paragraph and make the abstract one paragraph.

3. Page#2, Line#46: Replace “evaluating the proposed approaches' greenness” by “evaluating the greenness of the proposed approaches”
4. The purity of the studied drugs [troxerutin, dobesilate, and hydroquinine] should be written.
5. Under the procedure section, the statement "completing to the volume" must be changed to "completing to the mark."
6. Page #6 Line # 2: concentration in an ascending way.....should be deleted.
7. In section # 2.4.3.2.: The dosage form's nominal content was determined....., the obtained statistical data should be added in Table #4.
8. Page#7 Line #10: We used the higher sensitivity of the first approach..... Please rewrite the sentences in an acceptable scientific writing way.
9. Page#7, Line#42: Replace “Experimental condition optimization” by “Optimization of experimental conditions”.
10. Reference #14.....please, adhere to the journal guidelines for reference formatting.
11. The resolution and the shape of figure #1 are poor. Please, try to improve.
12. The number of figures is a large number and some of them can be moved to the supplementary material.

Author's Response to Decision Letter for (RSOS-201888.R0)

See Appendix A.

Decision letter (RSOS-201888.R1)

This year has been very difficult for everyone, and we want to take the opportunity to thank you for your continued support in 2020.

The Royal Society Open Science editorial office will be closed from the evening of Friday 18 December 2020 until Monday 4 January 2021. We will not be responding during this time. If you have received a deadline within this time period, please contact us as soon as possible to allow us to extend the deadline. If you receive any automated messages during this time asking you to meet a deadline, we offer apologies and invite you to respond after the festive period or during normal working hours.

With our best for a peaceful festive period and New Year, and we look forward to working with you in 2021.

Dear Dr Salim:

Title: Simultaneous estimation of troxerutin and calcium dobesilate in presence of the carcinogenic hydroquinone using green spectrofluorimetric method
Manuscript ID: RSOS-201888.R1

It is a pleasure to accept your manuscript in its current form for publication in Royal Society Open Science. The chemistry content of Royal Society Open Science is published in collaboration with the Royal Society of Chemistry.

RSC Associate Editor
Comments to the Author:
(There are no comments.)

Reviewer(s)' Comments to Author:

Appendix A

Response to Referees

Title: Green spectrofluorimetric estimation of troxerutin and dobesilate simultaneously in the presence of carcinogenic degradation product hydroquinone
Manuscript ID: RSOS-201888

Dear Editor-in-Chief,

On behalf of co-authors, I would like to thank the editorial board for giving us this chance to further revise our manuscript. We appreciate all the valuable comments raised by the editor and reviewers. All the comments have been considered in preparing the revised version of the manuscript and a point-by-point reply is represented below:

Reviewer 1:

1. The title can be better rewritten in a clearer way as follows: Simultaneous estimation of troxerutin and calcium dobesilate in presence of the carcinogenic degradation product hydroquinone using green spectrofluorimetric method.

Yes, the reviewer suggested title is more precise, concise, and representative. However, due to the journal limitations of title characters (150 ONLY), The title was rewritten more straightforwardly to be “Simultaneous estimation of troxerutin and calcium dobesilate in presence of the carcinogenic hydroquinone using green spectrofluorimetric method“

2. The manuscript includes some minor English mistakes and should be generally revised regarding grammatical or spelling mistakes.

The manuscript was revised carefully and corrected accordingly.

3. P2, L25: Replace “Due to the method's high simplicity and sensitivity” by “Due to the high method simplicity and sensitivity”.

“Due to the method's high simplicity and sensitivity” was replaced by “Due to the high method simplicity and sensitivity”.

4. Unify the use of the term “fluorimetry” instead of “fluorometry” all over the text.

The use of the term “fluorimetry” instead of “fluorometry” was unified all over the manuscript.

5. P2, L30: Replace “TROX ethanolic solution” by “ethanolic TROX solution”.
TROX ethanolic solution” was replaced by “ethanolic TROX solution”.
6. P2, L44: Replace “They were” by “The developed methods are”.
“They were” was replaced by “The developed methods are” accordingly.
7. Avoid writing details about the results at the introduction section.
Details about the results were removed in the introduction section starting from p2, L 49 up to P 3, L16.
8. Did the authors study the effect of surfactants in water or in ethanol?
The effect of surfactants was firstly studied in water, then in the presence of ethanol.
9. P7, L3: Replace “Results and the discussion” by “Results and discussion”.
“Results and the discussion” was replaced by “Results and discussion”.
10. In “Section 3. Results and discussion”, please give a highlight to the relation of the chemical structures of the studied analytes and the presence of their native fluorescence.
A highlight to the relation of the chemical structures of the studied analytes and the presence of their native fluorescence was written as being phenolic compounds in P7.
11. P8, L4: Please re-phrase this sentence: “All the mentioned attributes forced us to continue using ethanol as diluent”.
The sentence was re-phrased accordingly.
12. P11, L34: The sentence starting with “Considering the proposed” should be rewritten in a clear way.
The sentence starting with “Considering the proposed” was rewritten in a clear way accordingly.
13. The access dates in references 23, 26 and 27 should be updated and the URL addresses in reference 26 and 27 should be checked. Also, it is preferable to include reference to royal society open science journal.
- The access dates in references 23, 28 and 29 is updated and the URL addresses in reference 28 and 29 were checked.
- Two references to the royal society open science journal (25, 26) related to our study have been added.
14. References corresponding to the methods used for evaluation of greenness should be added to Table 6.

References corresponding to the methods used for the evaluation of greenness were added in Table 6 accordingly.

15. The manuscript lacks the figure numbers on the presented figures.
The figure numbers were added to the presented figures accordingly.

Reviewer: 2

1. One remark, the authors should give the values of the optical densities at the excitation and the emission wavelengths, used, to obtain excitation and emission spectra.

A paragraph representing the values of the optical densities of the studied compounds at the selected wavelengths was added in “*Results and discussion*” section P7. Moreover, the obtained absorption spectra of zero order were added in the “*Supplementary material*” file (Fig. S1, Page S2).

2. Second remark: They should write Fluorescence excitation and emission spectra.

The excitation and emission wavelengths were added to Figures No. 2,4,5 and 6.

Reviewer: 3

1. The language needs some revision and editing.

The language was revised and edited carefully.

2. Page#2, Line#32: Remove the space at the end of the paragraph and make the abstract one paragraph.

The space at the end of the paragraph was removed accordingly.

3. Page#2, Line#46: Replace “evaluating the proposed approaches' greenness” by “evaluating the greenness of the proposed approaches”

Evaluating the proposed approaches' greenness” was replaced by “evaluating the greenness of the proposed approaches”

4. The purity of the studied drugs [troxerutin, dobesilate, and hydroquinine] should be written.

The labeled purity of the studied drugs [troxerutin, dobesilate, and hydroquinine] was included in the text.

5. Under the procedure section, the statement "completing to the volume" must be changed to be "completing to the mark."

Under the procedure section, "completing to the volume" was changed to be "completing to the mark."

6. Page #6 Line # 2: concentration in an ascending way.....should be deleted.

Concentration in an ascending way..... the sentence was deleted.

7. In section # 2.4.3.2.: The dosage form's nominal content was determined....., the obtained statistical data should be added in Table #4.

The obtained statistical data (nominal content) was added in Table #4.

8. Page#7 Line #10: We used the higher sensitivity of the first approach..... Please rewrite the sentences in an acceptable scientific writing way.

We used the higher sensitivity of the first approach..... was rewritten in an acceptable scientific writing way.

9. Page#7, Line#42: Replace "Experimental condition optimization" by "Optimization of experimental conditions".

The "Experimental condition optimization" was replaced by "Optimization of experimental conditions".

10. Reference #14.....please, adhere to the journal guidelines for reference formatting.

Reference #14 was corrected according to the journal reference style.

11. The resolution and the shape of figure #1 are poor. Please, try to improve.

The resolution and the shape of figure #1 were improved accordingly.

12. The number of figures is a large number and some of them can be moved to the supplementary material.

Three figures were moved to the supplementary material.

Thanks in Advance.

Regards,

Mohamed M. Salim, Ph.D.